# Technical Note on Unilateral Biportal Lumbar Endoscopic Interbody Fusion

**Eugene Tze-Chun Lau † and Pang Hung Wu \*,†** 

Department of Orthopaedic Surgery, Juronghealth Campus, National University Health System, Singapore 609606, Singapore
* Correspondence: wupanghung@gmail.com
† Both authors contributed equally to this work.

**Abstract:** Unilateral biportal lumbar endoscopic interbody fusion is a relatively new technique in the field of minimally invasive spine surgery. It combines the benefits of preservation of the normal anatomy of the spine with direct visualization of the decompression of neural elements and endplate preparation for fusion. This results in high union rates and excellent outcomes for patients with back pain and lumbar spinal stenosis from spondylolisthesis while reducing the risk of injuries to the neural elements, endplate fractures and the theoretical rate of adjacent segment disease from disruption of the musculature. In this paper, we describe the steps and technical pearls pertaining to this technique and methods to avoid common pitfalls and complications. In conclusion, this technique would be a good tool in the armamentarium of a spinal surgeon specializing in minimally invasive spinal surgery.

**Keywords:** biportal endoscopic spine surgery; unilateral biportal lumbar endoscopic interbody fusion; degenerative spine disease; back pain; lumbar spinal stenosis; spondylolisthesis; spinal instability; minimally invasive spine surgery; spinal fusion



## 1. Introduction

Degenerative lumbar spinal conditions such as lumbar spinal stenosis and spondylolisthesis with elements of dynamic spinal instability benefit greatly from fusion surgeries [1,2]. However, traditional open methods of lumbar spinal fusion, such as the transforaminal lumbar interbody fusion (TLIF) technique described by Harms et al. [3], are often associated with increased morbidity and subsequent adjacent segment disease [4] as patients live longer. Previous studies comparing minimally invasive versus open interbody fusion showed a trend towards decreased risks of adjacent segment disease in minimally invasive techniques [5]. This was postulated to be due to the disruption of the normal anatomy and the musculature of the lumbar spine.

Hence, there is a growing interest in minimally invasive techniques to help prevent long term complications from adjacent segment disease as well as improved patient outcomes from reduced post-operative analgesia requirements, reduced post-operative transfusion requirements, reduced duration of hospitalization and quicker return to baseline functional levels [6,7]. Previous minimally invasive techniques for fusion described includes microscopic tubular technique transforaminal lumbar interbody fusion (MT-TLIF), anterior lumbar interbody fusion (ALIF), lateral lumbar interbody fusion (LLIF) and oblique lumbar interbody fusion (OLIF). The advent of endoscopic surgery brought on ever less invasive techniques such as the previously described uniportal endoscopic trans-Kambin lumbar interbody fusion (ETKLIF) [8] and uniportal endoscopic facet-sacrificing posterolateral transforaminal lumbar interbody fusion (EPTLIF) [9].

The endoscopic uniportal technique has a high learning curve [10] and often requires the use of modified or specialized equipment for surgery. Biportal techniques aim to

overcome these unique challenges through the addition of a second incision, allowing the use of more conventional spine surgical instruments similar to open surgery. This results in a lower learning curve and quicker transference of open spinal surgery skill sets to endoscopic surgery. It also allowed greater maneuverability and new angles for approach and access to the spine.

In this paper, we present our technique for the unilateral biportal lumbar endoscopic interbody fusion technique and outline the common pitfalls and complications associated with it.

## 2. Surgical Anatomy

Understanding the anatomy of the Kambin's triangle and the corridor of approach is essential during navigation for endoscopic spine procedures. The Kambin's triangle [11] is defined as a right-angled triangle with the superior endplate of the caudal vertebra as the horizontal leg, the lateral dural edge of the cauda equina as the vertical leg and the caudal border of the exiting nerve root as the hypotenuse (Figure 1). This is the common corridor of safety used for access to the disc in fusion surgeries utilized in posterior approaches—be they open or minimally invasive.

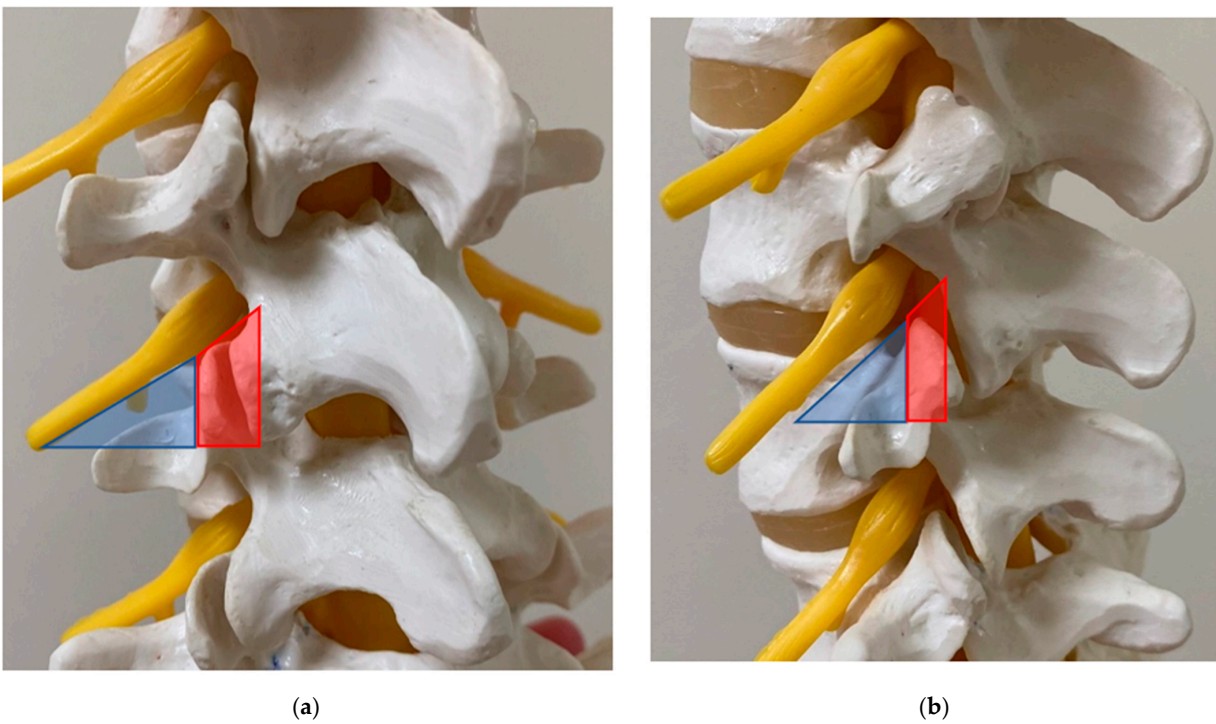

      (**a**)                                       (**b**)

**Figure 1.** Posterior oblique (**a**) and lateral (**b**) views of the lumbar spine showing the medial (shaded red) and lateral (shaded blue) portions of the Kambin's triangle.

In the facet-preserving ETKLIF, only the lateral portion of the Kambin's triangle is utilized, with the medial portion blocked by the facet joints [8,9]. This smaller working space and corridor resulted in the need for modified equipment, limitation in size of interbody cages used as well as increases the risks of injury to the exiting nerve root in the hands of an inexperienced surgeon.

Similar to open and EPTLIF, our unilateral biportal lumbar endoscopic interbody fusion technique utilizes the medial portion of the Kambin's triangle via facetectomy of the inferior articular process (IAP) of the cranial vertebra and superior articular process (SAP) of the caudal vertebra [9]. This method creates a bigger working space to allow for better decompression and visualization of the neural elements, as well as a larger corridor of access for discectomy and endplate preparation.

To perform the facetectomy, it is important to identify two endoscopic landmarks previously described [12]—Kim's point and Wu's point (Figure 2) Kim's point is defined as the point of intersection of the superolateral border of the cranial IAP with the superolateral border of the caudal SAP, marking the lateral limit of the facetectomy. On the other hand, Wu's point is defined as the point of intersection of the superomedial border of the cranial IAP with the medial edge of the cranial lamina overlapping the superomedial aspect of the caudal SAP, marking the medial limit of the facetectomy. After identification of these two points during endoscopy, a bone drill can be used for resection of the cranial IAP from Wu's point to Kim's point. Osteotomy along this line is safe as the SAP protects the underlying neural structures. This allow access to the medial portion of the Kambin's triangle underneath.

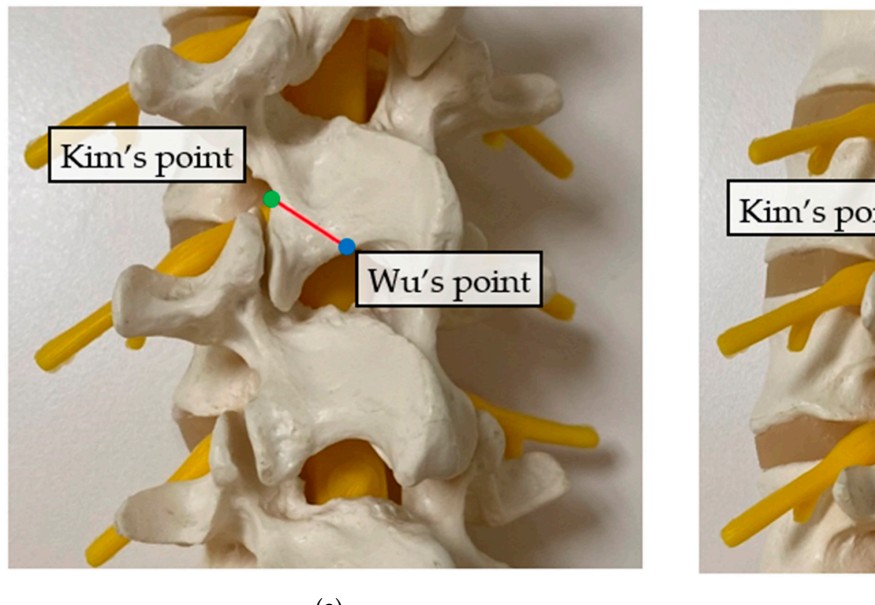
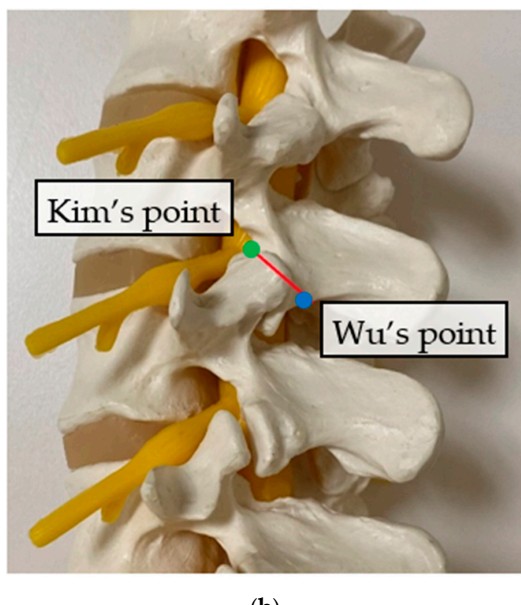

|         (**a**)         |         (**b**)         |

**Figure 2.** Posterior oblique (**a**) and lateral (**b**) views of the lumbar spine showing Kim's point (green dot), Wu's point (blue dot) and the line of osteotomy of IAP (red line).

### 3. Surgical Technique

*3.1. Anaesthesia and Positioning*

After the patient is placed under general anesthesia and neuromonitoring attached, the patient is positioned prone on a radiolucent table and supported at the chest and the pelvis, leaving the abdomen free. The table is then tilted to get the endplates desired perpendicular to the floor using fluroscopic guidance.

*3.2. Equipment*

An arthroscopic system with a 0° scope and continuous normal saline irrigation is essential. We would also recommend a corresponding biportal endoscopic spine surgery set (BESS™ set, MGB Endoscopy Co., Ltd., Seoul, Korea) containing Bonss radiofrequency plasma surgical systems for radiofrequency ablation (Bonss Medical®, Jiangsu, China) and a high-speed diamond burr as per the surgeon's preference. With regards to the interbody cages and screw systems, these can also be left to the surgeon's preference.

*3.3. Skin Incision and Docking*

Fluoroscopy is used to mark out the mid pedicle line. For right-hand dominant surgeons, a left-sided approach is used. The first 5–6 mm incision is made for the smaller viewing portal at the level of the inferior border of the pedicle of the cranial vertebra. A

second 8–10 mm incision is made for the larger instrument portal to also allow outflow of irrigation. The two incisions should be separated by 2–3 cm.

Serial dilators up to 10 mm are used to split the paraspinal muscles and a periosteal elevator is used to gently detach the soft tissue off the interlaminar space (in the lateral-to-medial direction). A sleeve system can be used to maintain the muscle split and prevent injury to the paraspinal muscles. The inflow of normal saline helps to create and maintain a submuscular working space in addition to providing a hydrostatic pressure to assist hemostasis. A radiofrequency wand is used to clear soft tissue allowing visualization of the anatomy and to cauterize bleeding. We utilize the inside-out approach (docking and starting from Wu's point and working towards Kim's point). After identifying the anatomy, the goal of the next step is triangulation and docking of the arthroscope and the working instrument onto Wu's point, as described earlier under surgical anatomy.

### 3.4. Inferior Articular Process Facetectomy

We recommend performing the facetectomy prior to decompression and removal of the intervening ligamentum flavum. We start by using a high-speed burr to mark out both Wu's point and Kim's point and to thin out the bone in a line connecting both points. We then use an osteotome to complete the osteotomy, allowing harvesting of the IAP for use as local bone graft (Figure 3). This subsequently exposes the medial aspect of the SAP.

### 3.5. Superior Articular Process Facetectomy

We use the high-speed burr with assistance of a 90° Kerrison punch to mark out and remove the medial aspect of the base of the SAP where it connects to the inferior lamina. The medial half of the SAP is then harvested with an osteotome for use as bone graft as well (Figure 4). It is important to ensure that sufficient base of SAP is removed to allow a smooth corridor for access to the disc.

### 3.6. Decompression and Discectomy

Decompression of the lumbar spinal canal can be performed using a high-speed diamond burr or a traditional Kerrison punch for an ipsilateral laminotomy with an over-the-top decompression of the contralateral lateral recess. After removal of the deep ligamentum flavum, the neural elements should be fully visible and the Kambin's triangle in the axilla of the exiting nerve root identified. We use the radiofrequency probe for diathermy of dural veins and annulutomy. Discectomy can then be performed through this safe zone using conventional pituitary and curettes (Figure 5).

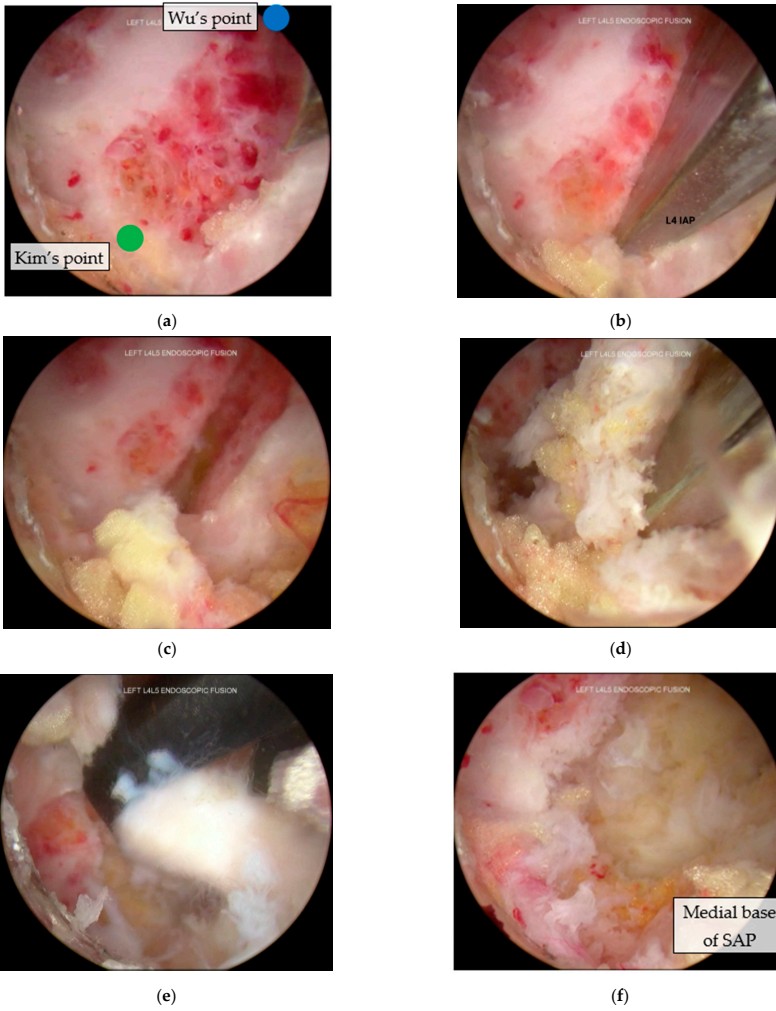

**Figure 3.** (**a**)—A high-speed burr was used to mark both Wu's point (blue dot) and Kim's point (green dot) and to thin out the bone in a line connecting both points. (**b**)—Completion of osteotomy using an osteotome. (**c**)—Showing the depth of osteotomy, completely through the IAP until reaching the SAP. (**d**)—Levering out the osteotomized IAP using the osteotome. (**e**)—Removal of the IAP fragment using a Kerrison punch. (**f**)—After removal of IAP, leaving behind the underlying ligamentum flavum and the medial aspect of SAP.

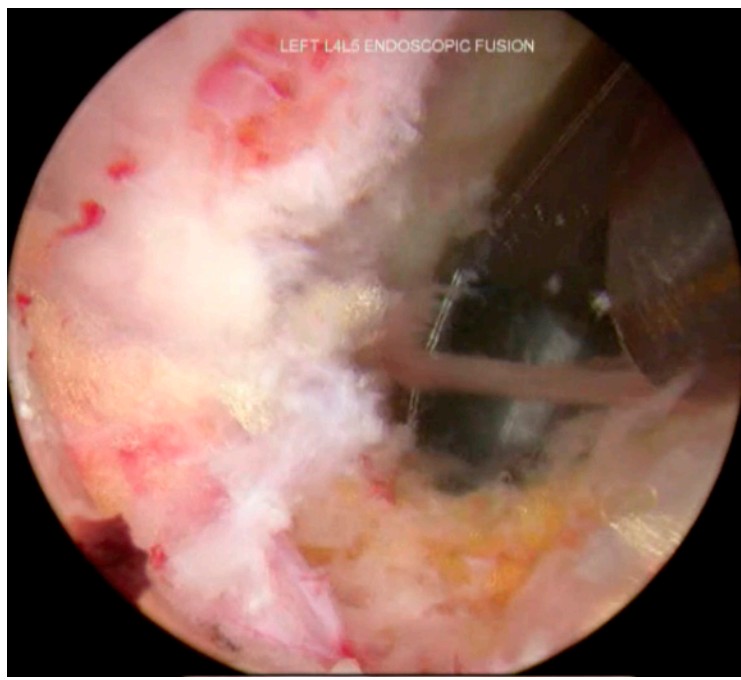

**Figure 4.** The medial aspect of the base of the SAP was rongeured off using a 90° Kerrison punch, allowing the use of an osteotome to complete the removal of SAP.

### 3.7. Endplate Preparation

After discectomy, we maneuver the endoscope into the disc space to allow direct visualization of the endplate. An endoscopic penfeel is used to remove the remaining cartilaginous endplate after the initial scraping by a curette (Figure 6). This ensures complete removal of the cartilaginous endplate and a better bed for fusion. Furthermore, as we are able to minimize the forces required by removing the cartilaginous endplate using a penfeel instead of scraping with a curette, we reduce the risk of endplate fractures. Bone graft harvested earlier from the SAP and IAP is morselized and packed into the disc space under direct vision to ensure symmetrical distribution and prevent overstuffing.

### 3.8. Cage Insertion with Retractors

At this point, there is usually sufficient working space within the Kambin's triangle for insertion of the interbody cage. However, if there is a risk of injury to the neural elements, a modified nerve root retractor can be inserted and held by an assistant under direct vision. Sequential dilation and cage sizing can be performed under fluoroscopy (Figure 7). The cage can be packed with bone graft or augmented with demineralized bone matrix or allografts to promote fusion as per standard open fusion preference.

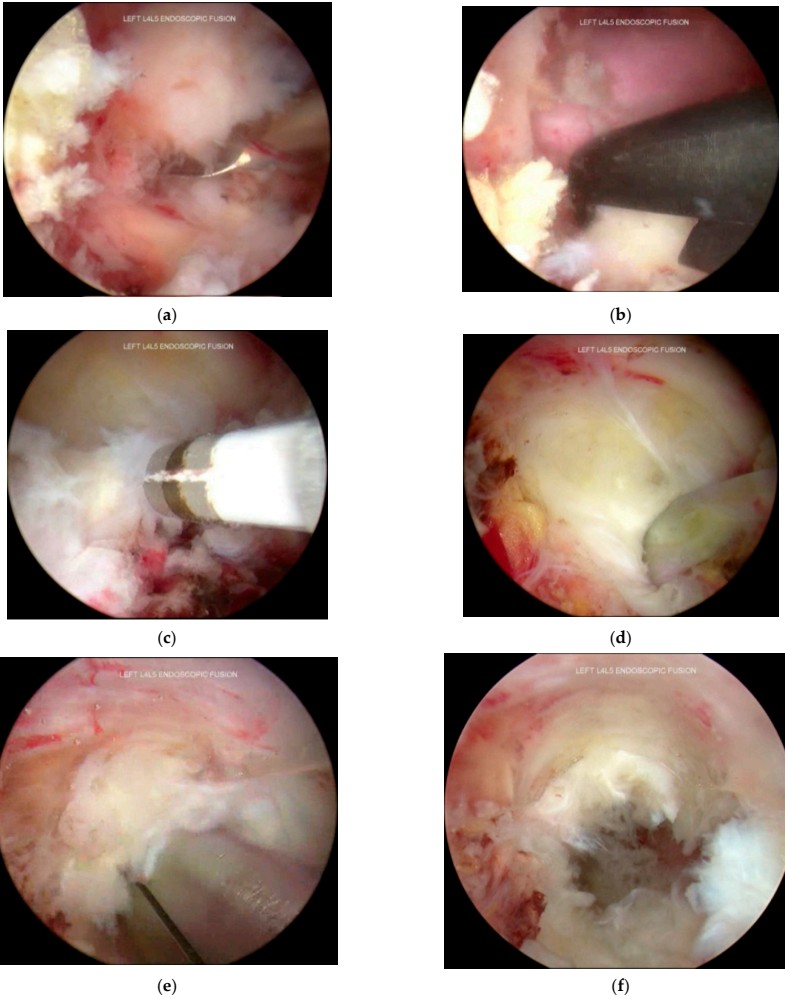

**Figure 5.** (**a**)—Splitting of the ligamentum flavum and development of the interval between the underside of the deep ligamentum flavum and the dura using a curette. (**b**)—Removal of ligamentum flavum after detaching them from the dura using a combination of Kerrison punch and pituitary forceps. (**c**)—Diathermy of dural veins using a radiofrequency probe. (**d**)—Identification of the disc and annulotomy. (**e**)—Discectomy using pituitary and curettes. (**f**)—Protection of neural elements using the camera sleeve.

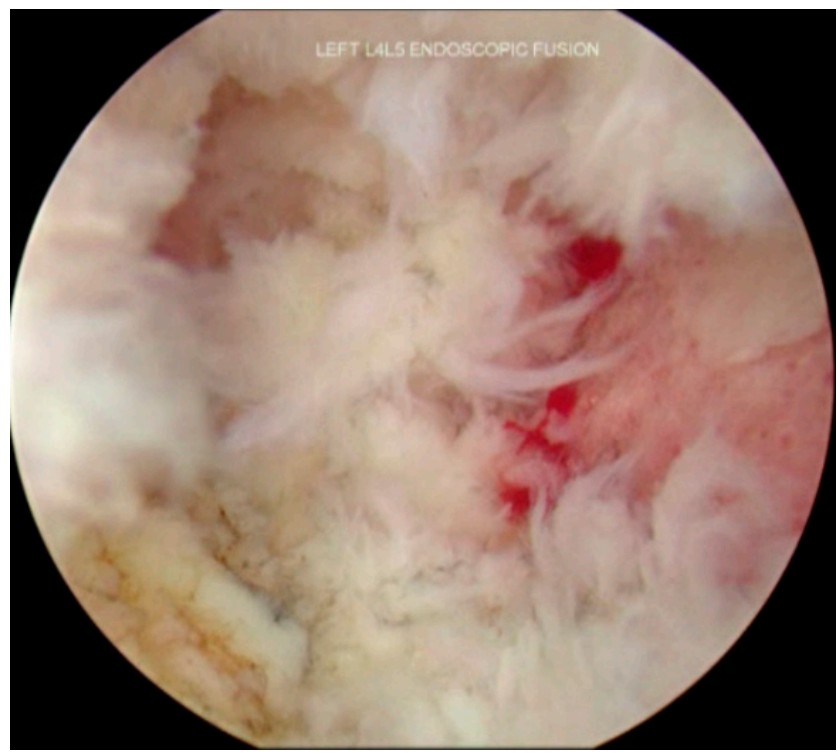

**Figure 6.** Visualization of endplate preparation and remaining cartilage.

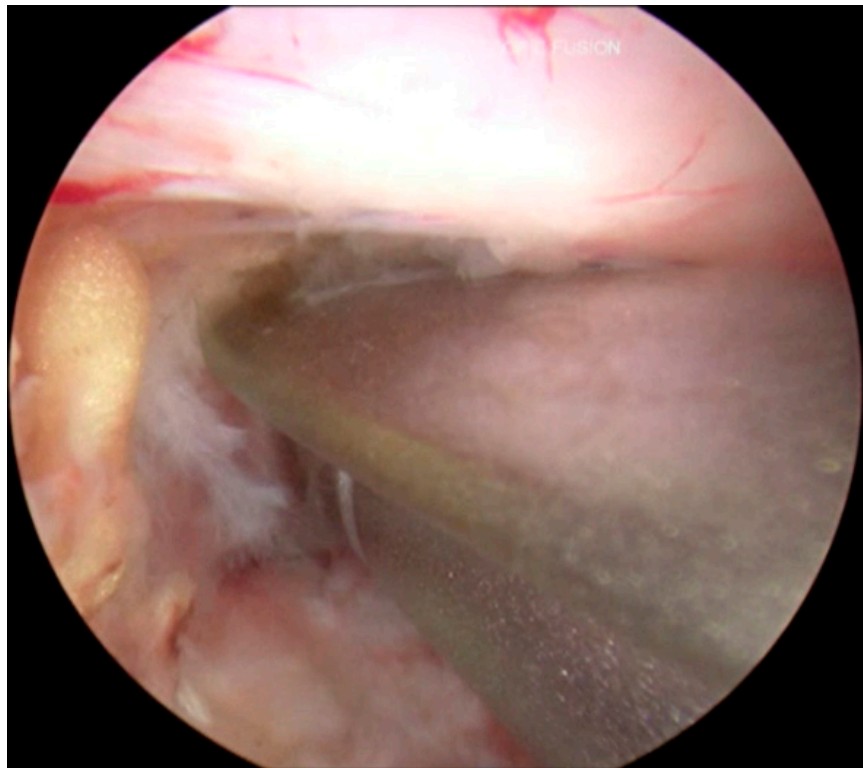

**Figure 7.** Use of a nerve root retractor to protect the neural elements during sizing and insertion of the interbody cage.

*3.9. Percutaneous Posterior Instrumentation under Fluoroscopic Guidance*

After confirmation of the cage position with fluoroscopy together with endoscopy, pedicle screws can be inserted percutaneously under fluoroscopic guidance. If the bed was flexed earlier, it is important to remember to correct it to create the needed lumbar lordosis prior to insertion of the rods and finalizing the construct.

## 4. Case Example

*4.1. Case 1: Madam M*

Madam M was a 67-year-old lady with a two-year history of back pain with bilateral lower limb radiculopathy (bilateral posterior thigh and posterior calf pain) and claudication time of 30 min. She had weakness of her left extensor hallucis longus (L5 myotome)—grade 4. She had no numbness in her lower limbs. Radiographs of the lumbar spine showed L3/L4 decreased disc height (Figures 8 and 9). MRI scan of the lumbar spine showed L3/L4 degenerative disc disease with bilateral lateral recess stenosis at the same level (Figure 10). To confirm the source of her pain, she underwent L3/L4 discogram that confirmed that as the source of her pain. She subsequently underwent a left endoscopic L3/L4 fusion. She was discharged on the first post-operative day. During clinic review at 2 weeks, 3 months and 10 months, her back pain and her bilateral lower limb radiculopathy had improved gradually until it completely resolved at the most recent review at 10 months (Figure 11).

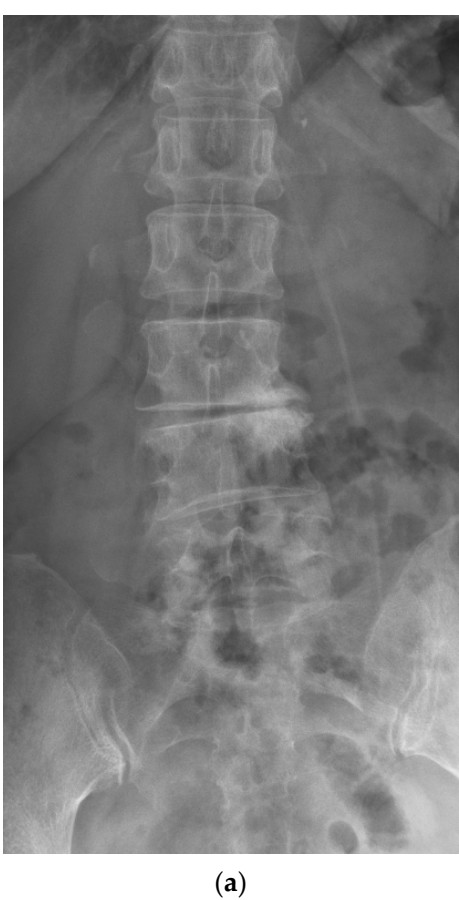
**(a)**

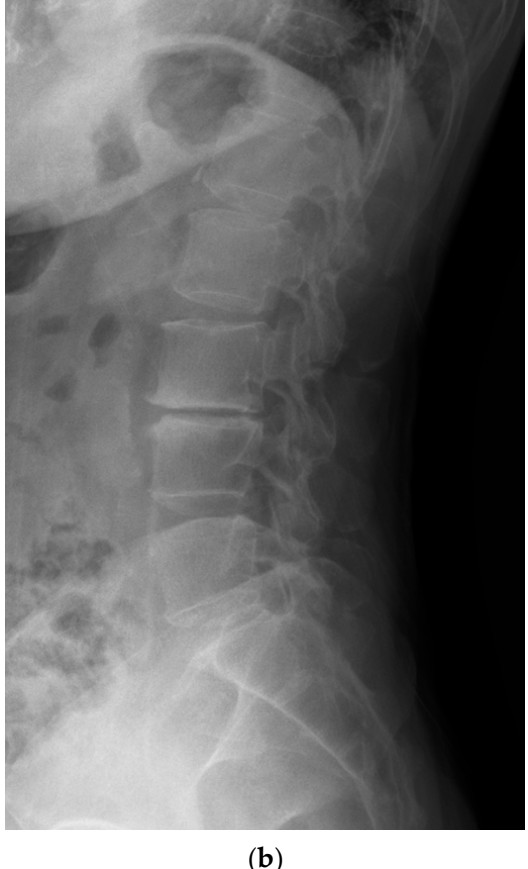
**(b)**

**Figure 8.** (**a**,**b**)—AP and lateral radiographs of the lumbar spine showing L3/L4 decreased disc height.

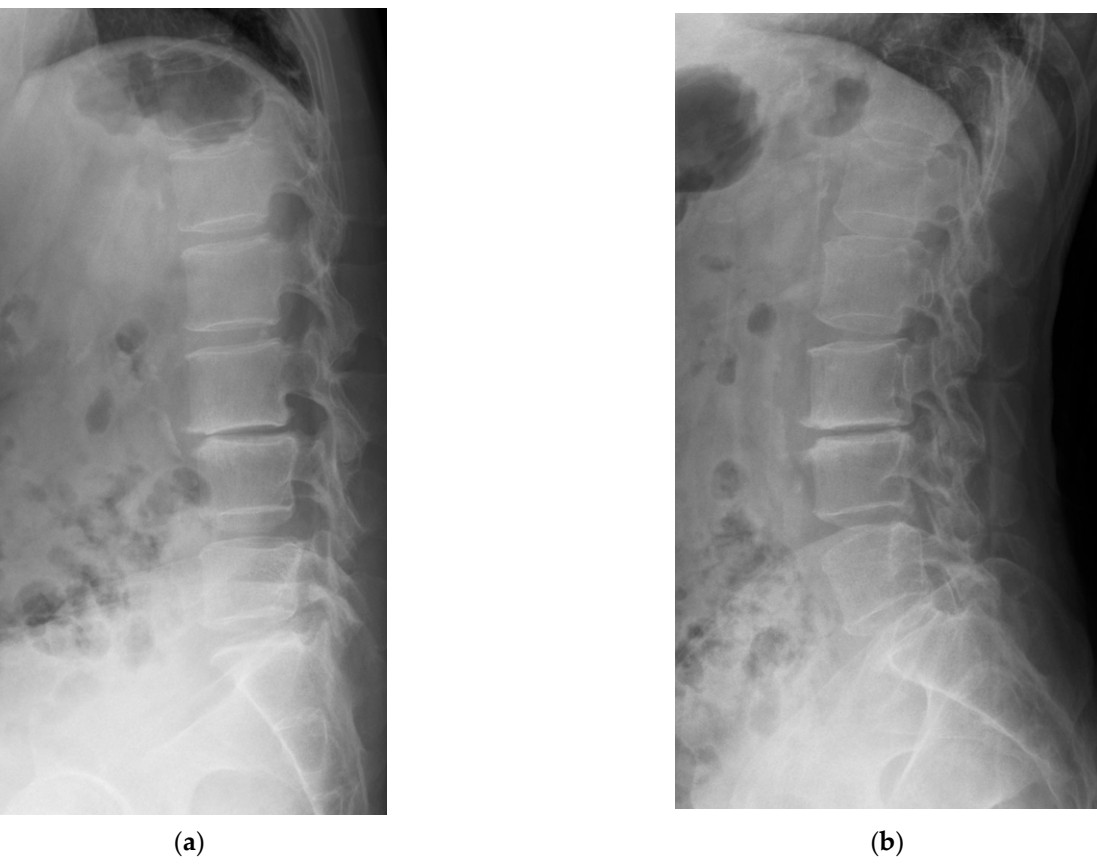

(**a**)  (**b**)

**Figure 9.** (**a**,**b**)—Flexion and extension radiographs of the lumbar spine showing slight dynamic instability at L3/L4.

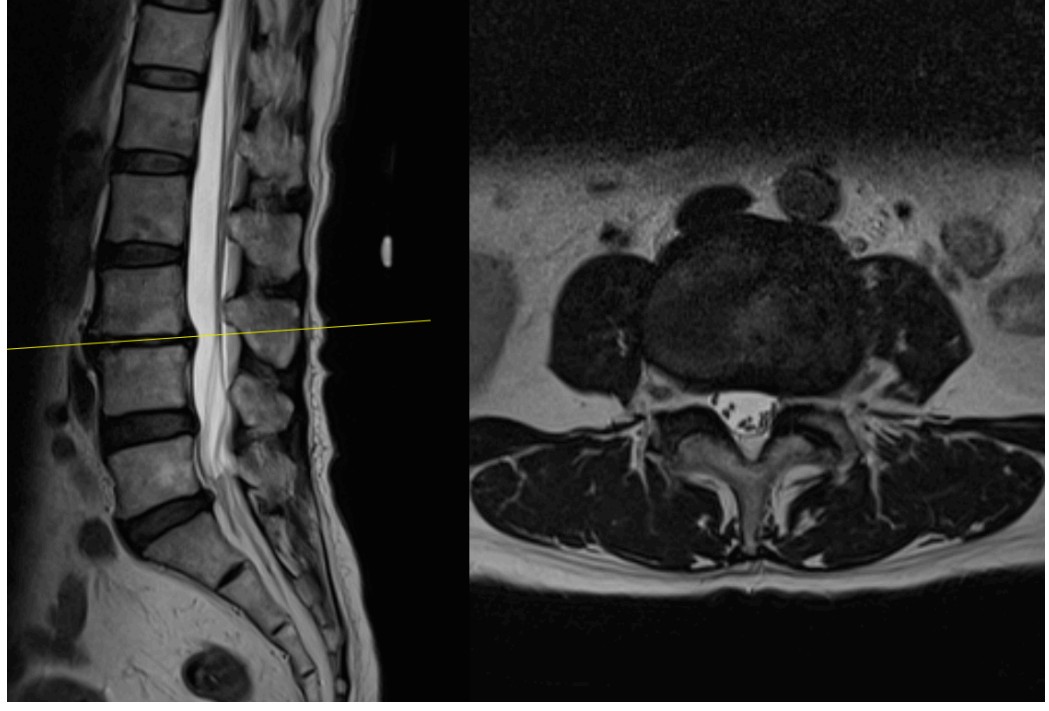

**Figure 10.** MRI scan of the lumbar spine showing L3/L4 degenerative disc disease with bilateral lateral recess stenosis at the same level (left more than right).

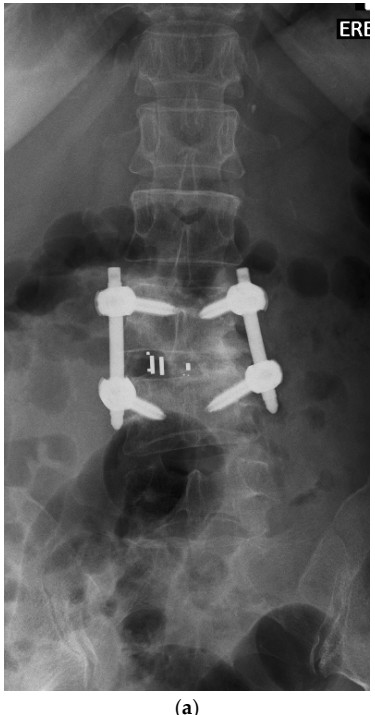
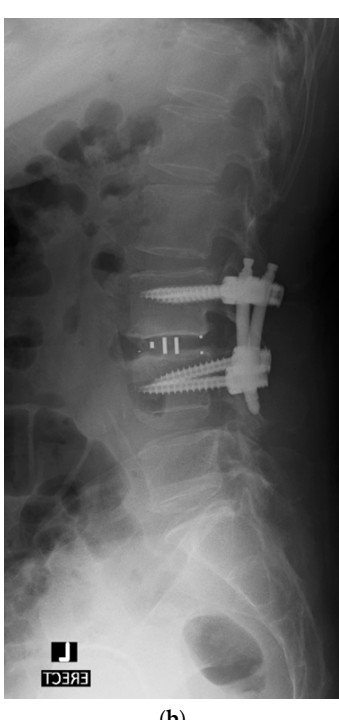

(**a**)  (**b**)

**Figure 11.** (**a**,**b**)—AP and lateral radiographs of the lumbar spine at 3 months post-operatively.

*4.2. Case 2: Madam R*

Madam R was a 56-year-old lady with back pain and bilateral calf claudication of approximately 500 m distance. She had no weakness or numbness in either of her lower limbs. Radiographs of the lumbar spine showed L4/L5 grade 2 spondylolisthesis (Figures 12 and 13). MRI scan of the lumbar spine showed lumbar spinal stenosis at L4/L5 level (Figure 14). She underwent a left endoscopic L4/L5 fusion (Figure 15). She was discharged on the second post-operative day. During clinic review at 2 weeks and 3 months, she had no back pain and her claudication symptoms had completely resolved.

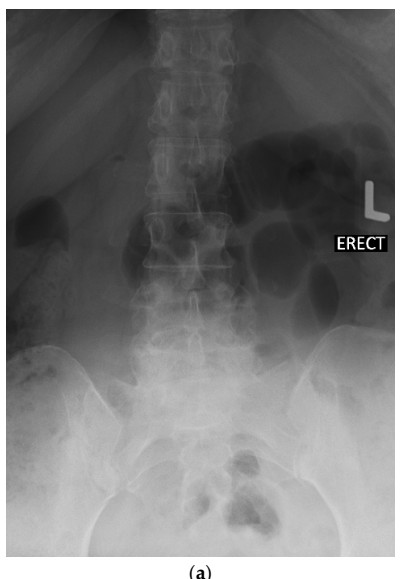
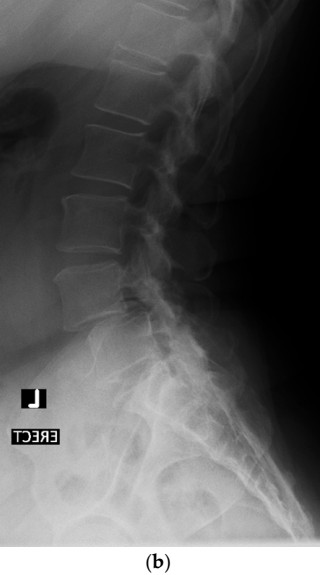

(**a**)  (**b**)

**Figure 12.** (**a**,**b**)—AP and lateral radiographs of the lumbar spine showing L4/L5 grade 2 spondylolisthesis.

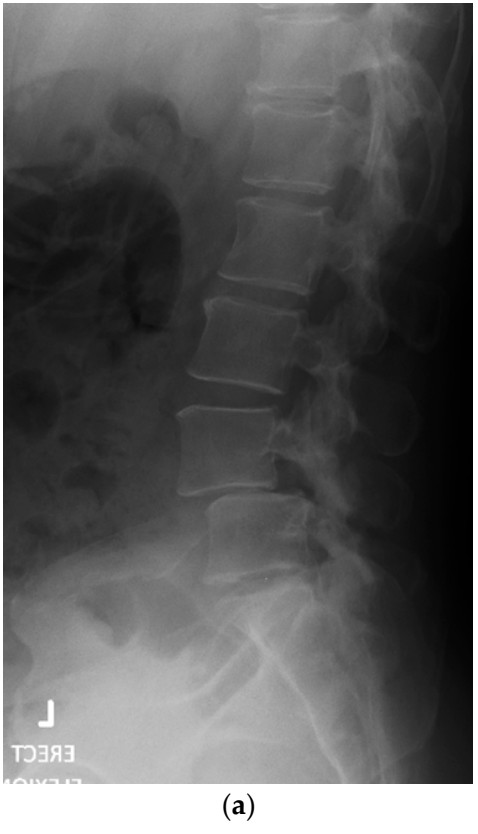 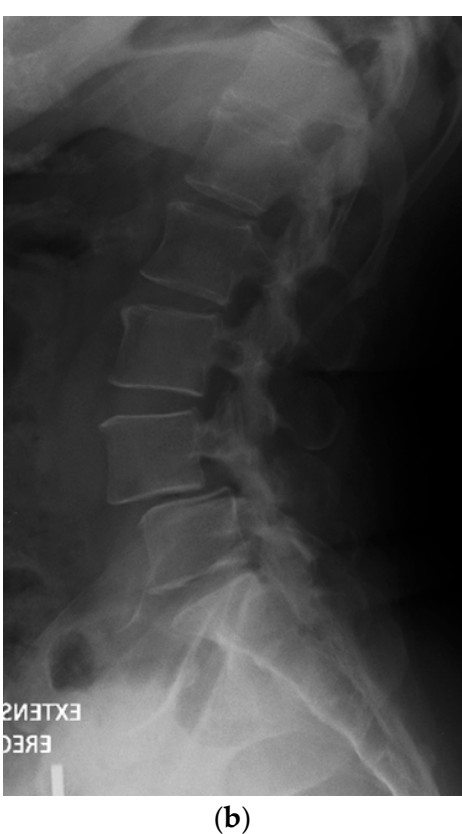

(**a**) (**b**)

**Figure 13.** (**a**,**b**)—Flexion and extension radiographs of the lumbar spine did not show any significant dynamic instability at L4/L5.

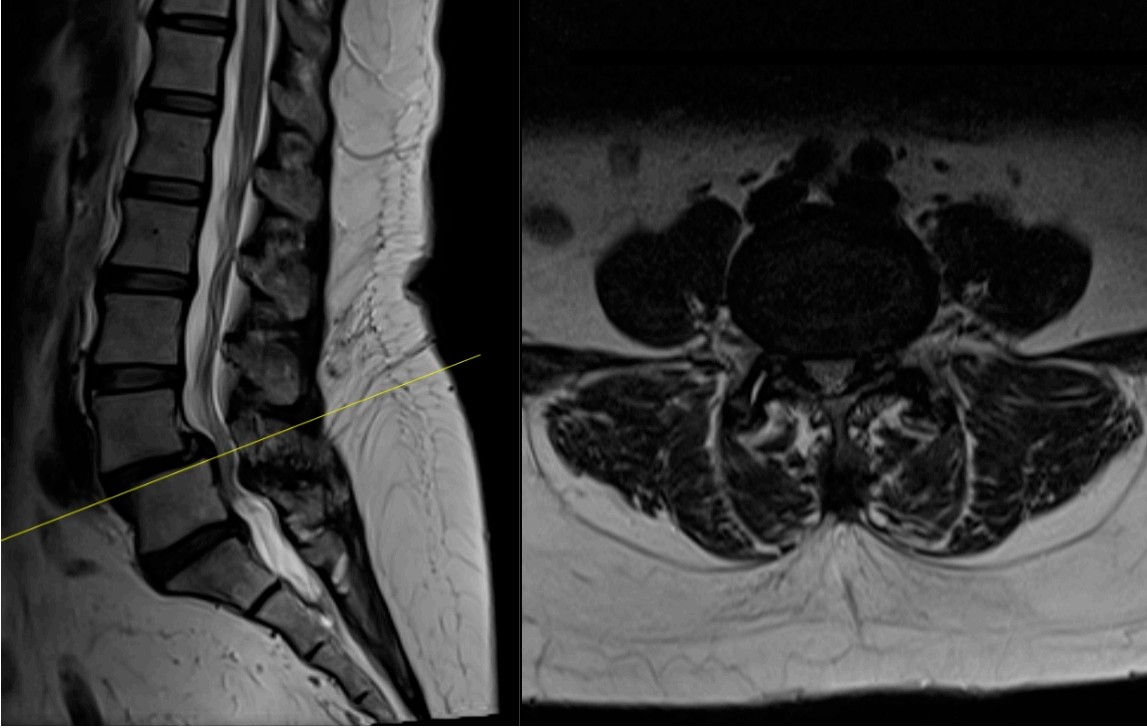

**Figure 14.** MRI scan of the lumbar spine showing L4/L5 spondylolisthesis with lumbar spinal stenosis at the same level.

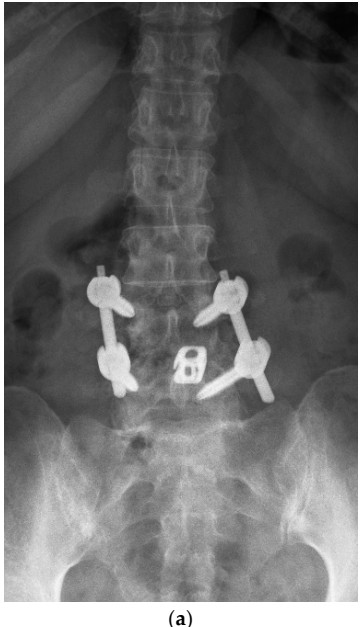
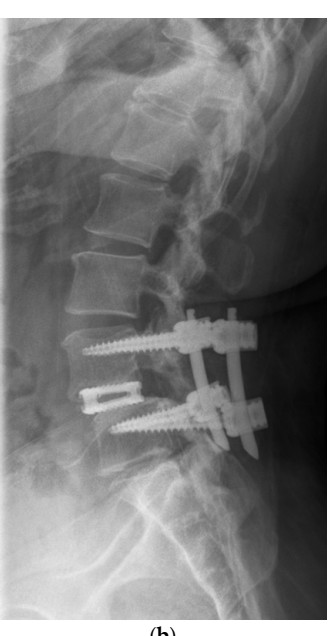

(**a**)                                                                                  (**b**)

**Figure 15.** (**a**,**b**)—AP and lateral radiographs of the lumbar spine at 3 months post-operatively.

*4.3. Case 3: Madam M*

Madam M was a 64-year-old lady with back pain with bilateral lower limb radiculopathy (bilateral posterior thigh and posterior calf pain). She had no weakness in both her lower limbs but she had reduced sensation in her right L5 and S1 dermatomes. Radiographs of the lumbar spine showed L3/L4 grade 1 spondylolisthesis, L4/L5 and L5/S1 decreased disc height (Figures 16 and 17). MRI scan of the lumbar spine showed L3/L4 and L4/L5 lumbar spinal stenosis (Figure 18). She underwent a left endoscopic L3 to S1 fusion (Figures 19 and 20). She was discharged on the second post-operative day. During clinic review at 2 weeks, her back pain and her bilateral lower limb radiculopathy had improved.

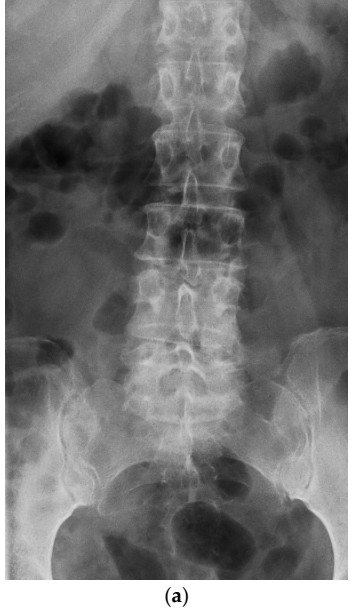
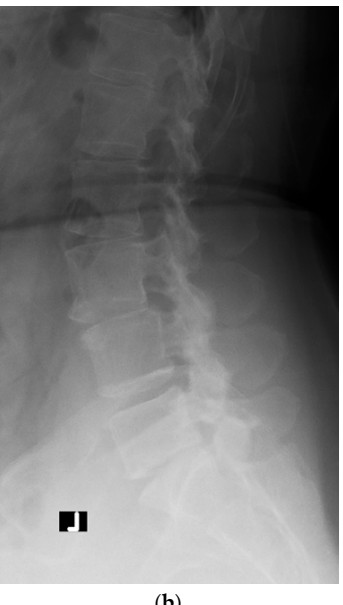

(**a**)                                                                                  (**b**)

**Figure 16.** (**a**,**b**)—AP and lateral radiographs of the lumbar spine showing grade 1 L3/L4 spondylolisthesis, and L5/S1 decreased disc height.

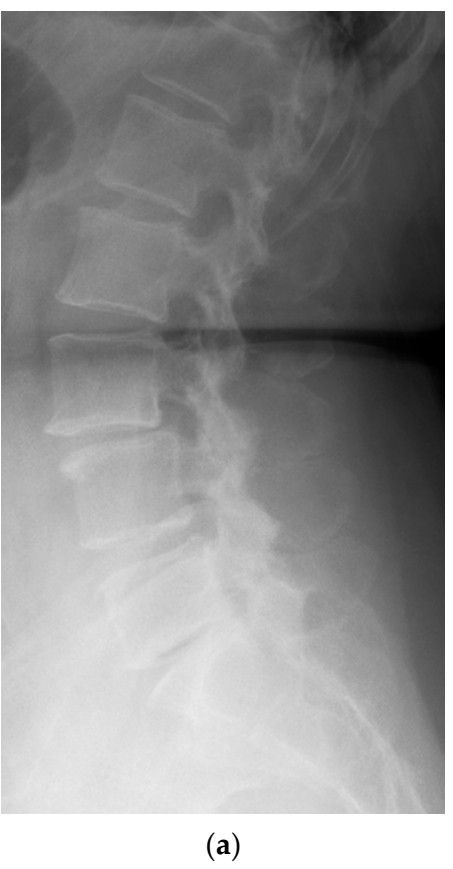

(**a**)

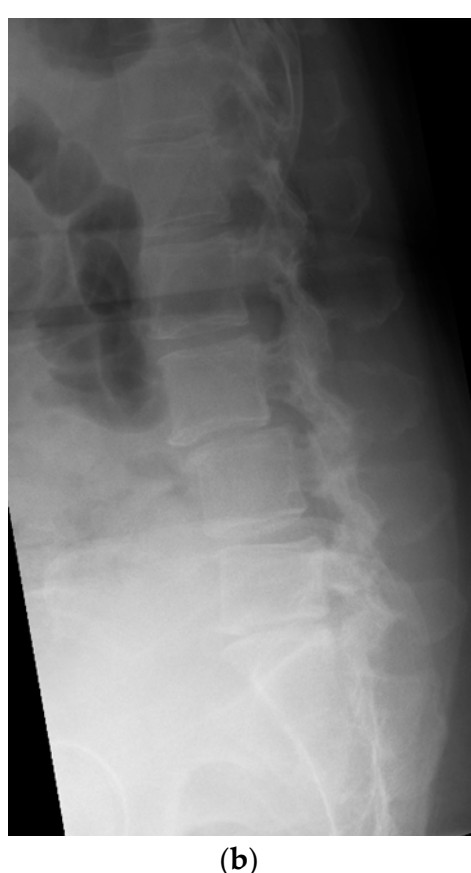

(**b**)

**Figure 17.** (**a**,**b**)—Flexion and extension radiographs of the lumbar spine did not show any significant dynamic instability at L3/L4.

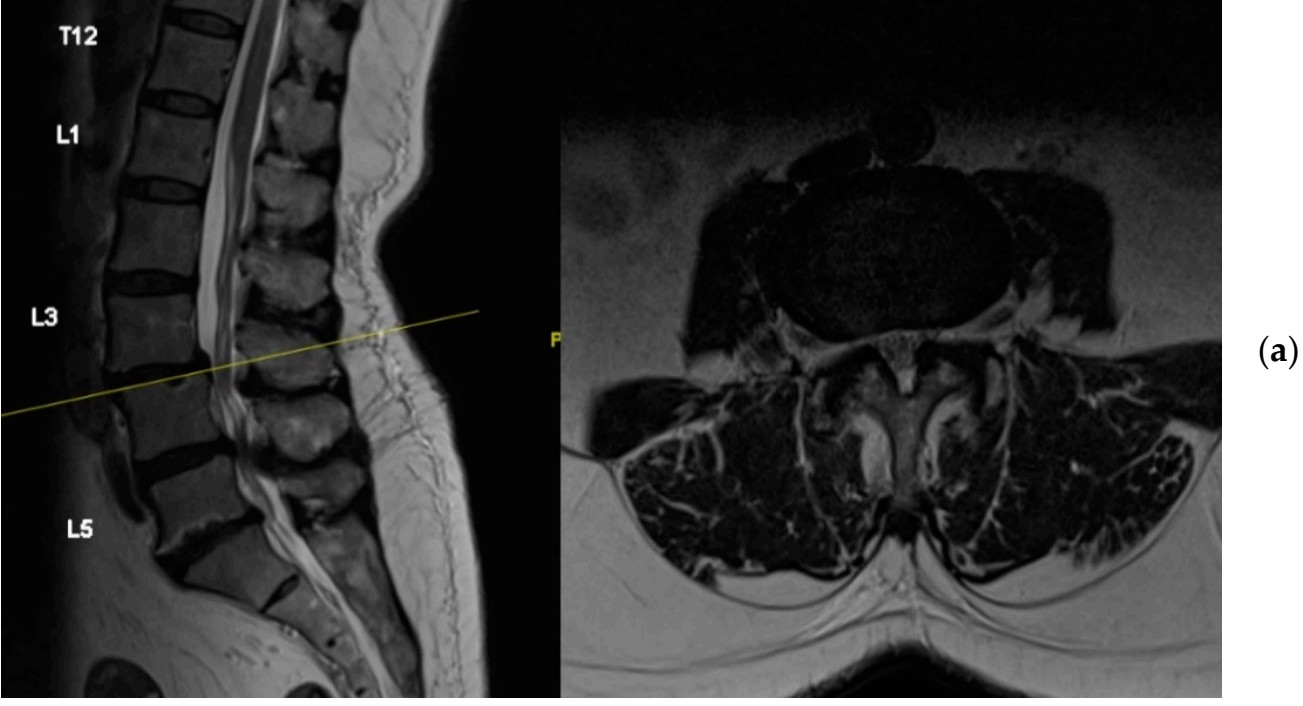

(**a**)

**Figure 18.** *Cont*.

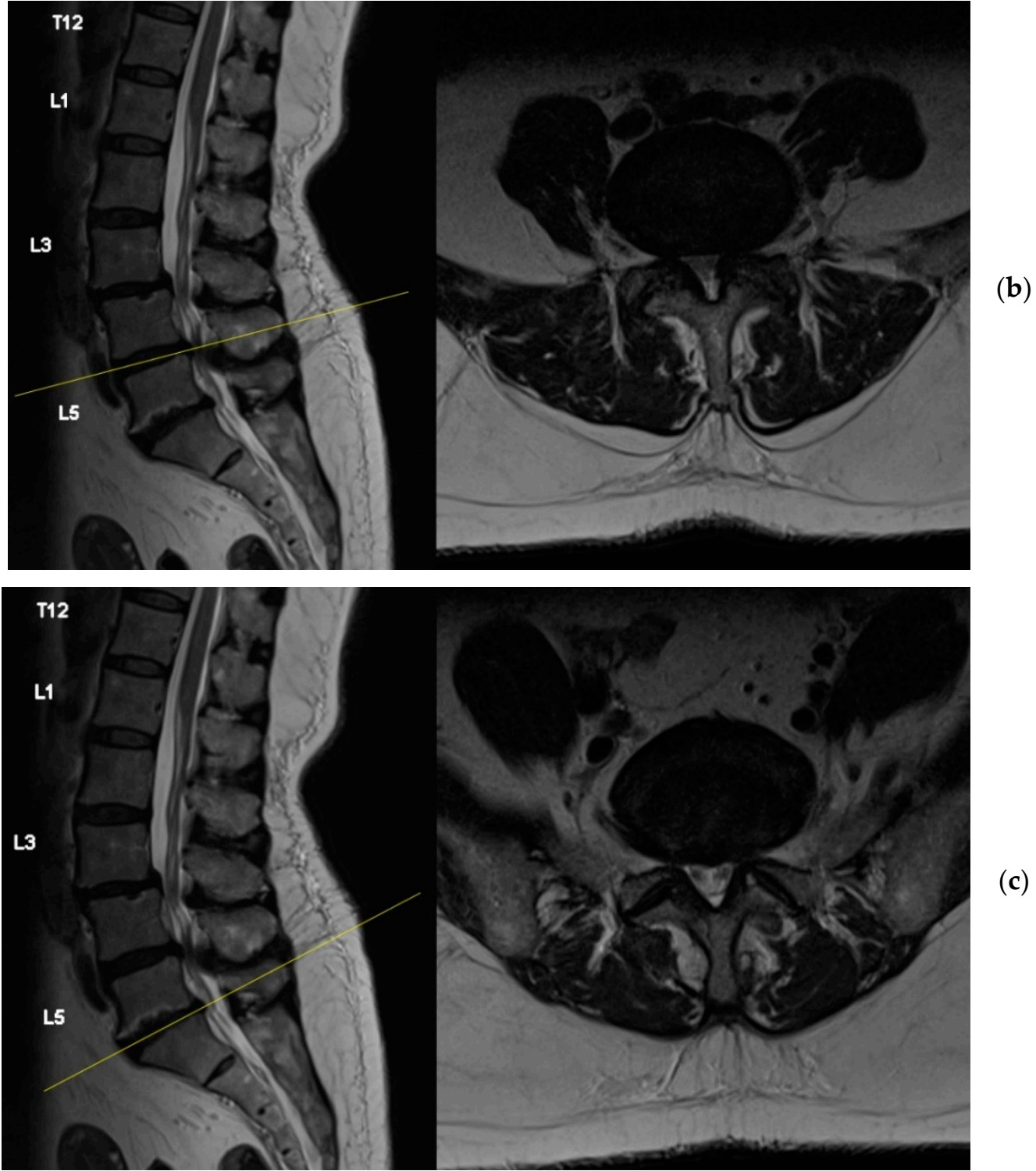

**Figure 18.** MRI scan of the lumbar spine showing (**a**) L3/L4 spondylolisthesis with bilateral lateral recess stenosis at the same level, (**b**) L4/L5 lumbar spinal stenosis and (**c**) L5/S1 degenerative disc disease with a disc bulge and bilateral lateral recess stenosis.

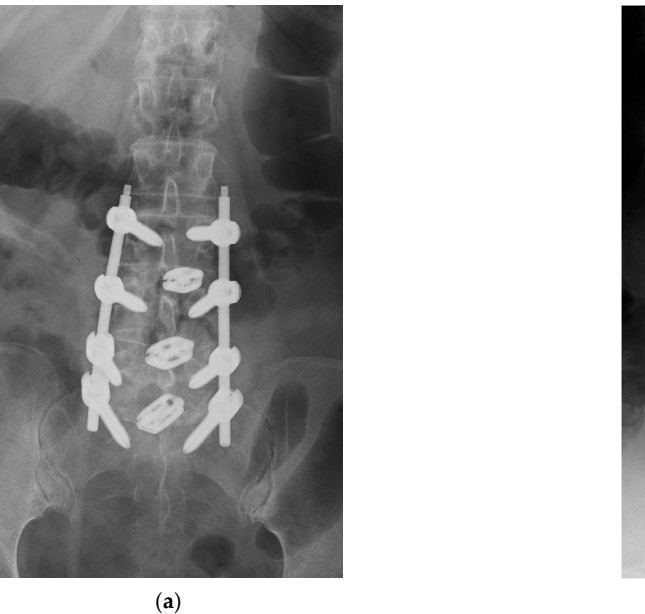

(**a**) (**b**)

**Figure 19.** (**a**,**b**)—AP and lateral radiographs of the lumbar spine at 2 weeks post-operatively.

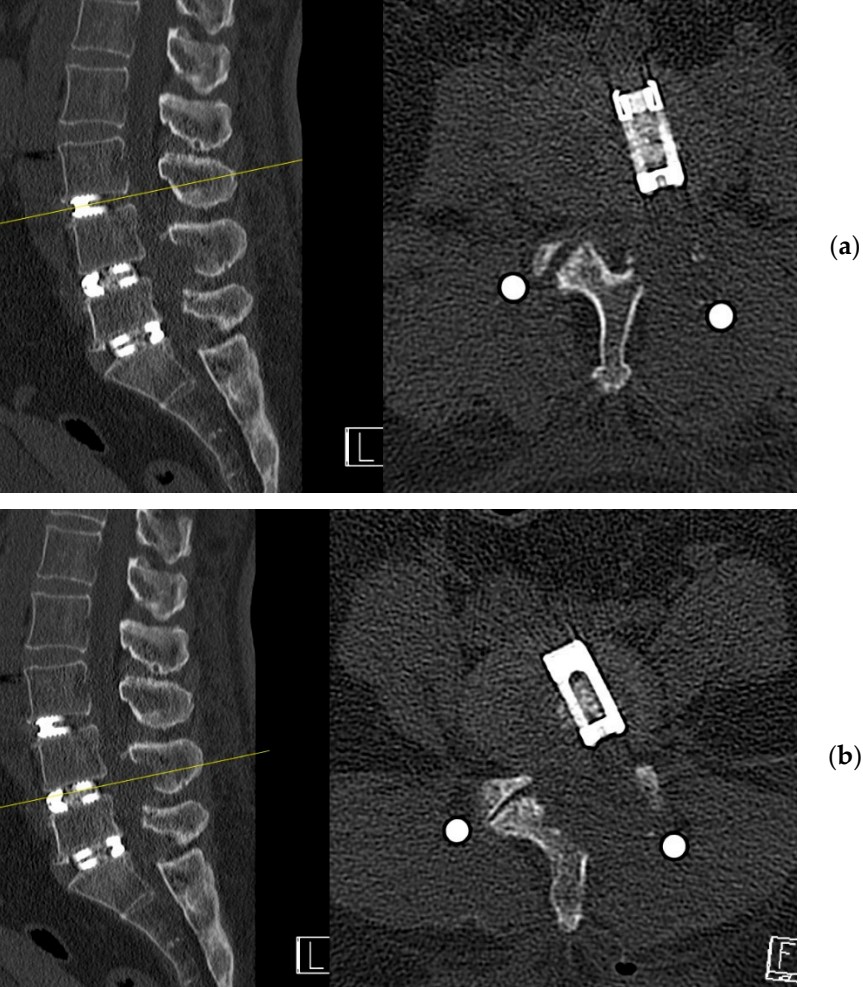

(**a**)

(**b**)

**Figure 20.** *Cont*.

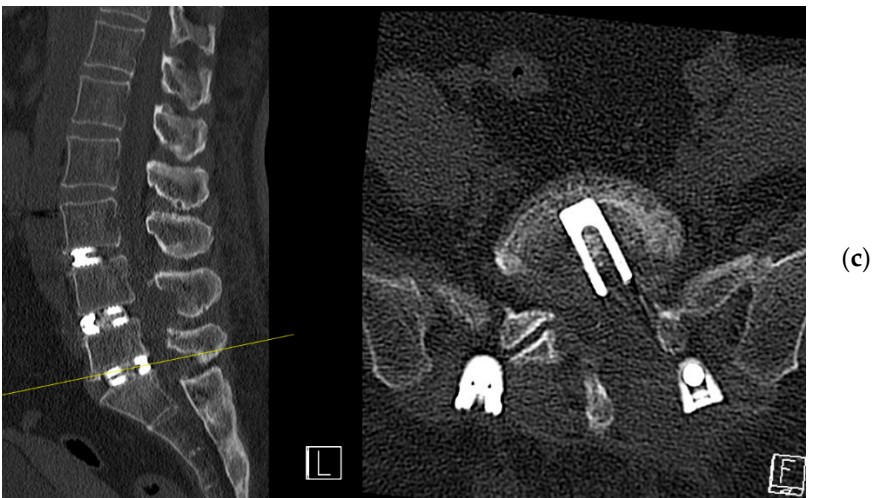

**Figure 20.** CT scan of the lumbar spine showing the amount of laminotomy/bone removed at (**a**) L3/L4, (**b**) L4/L5 and (**c**) and L5/S1.

## 5. Discussion

Endoscopic fusion techniques are relatively new compared to the more established open techniques. Many senior surgeons are also previously trained and more familiar with open techniques. This results in a reluctance to convert to using endoscopic techniques due to the high learning curve associated with them [10,13]. This is particularly true for uniportal endoscopic surgeries and trans-Kambin facet-preserving approaches. The smaller working space especially in cases with spondylolisthesis, in addition to the limitations in the size of cages that can be used without causing risks of injury to the neural elements, often pose a high entry bar for prospective surgeons.

Our technique for endoscopic fusion presented here allows the use of larger interbody cage sizes as we remove the facet completely for use as local bone graft anterior to the cages. Larger cages and expandable cages can help to deformity correction and create lumbar lordosis to improve the patient's sagittal balance. Larger oblique lumbar interbody fusion cages can also be used if the distance between the exiting and traversing nerve is more than 13mm. In our case examples, we showed the use of both standard interbody cages (in case 1) and expandable cages (in cases 2 and 3). We routinely use expandable cages to help achieve more lumbar lordosis. However (as in case 1), we recommend the use of standard interbody cages in patients with osteoporosis (with a bone mineral density scan T-score of $<-2.5$) to avoid endplate injuries and prevent subsidence.

The development of unilateral biportal techniques for interbody fusion would help to bridge this gap as it allows a lower learning curve and quicker transference of open spinal surgery skill sets to endoscopic surgery. It also utilizes the same corridor as the established open TLIF and MT-TLIF with similar sets of equipment, albeit with the use of endoscopy instead of a microscope. Surgeons who are familiar with the triangulation during arthroscopy utilized in other joints would also be able to pick up endoscopic biportal techniques more easily. This would allow greater traction and acceptance of endoscopic spinal surgeries in the surgical community, resulting in further understanding and development of these techniques.

However, unilateral biportal techniques does face its own set of unique challenges. Due to the biportal nature of the technique, surgeons need to become familiar with using the non-dominant hand for holding the endoscope, allowing only the dominant hand for handling instruments for decompression. An assistant may also be required for hammering during osteotomy coupled with holding the retractor when working within the disc space. As with all endoscopic techniques, patients with coagulopathies also make visualization difficult due to the need for meticulous hemostasis. There is also a risk related to the use of irrigation fluid. It is important to maintain a low irrigation pressure after the

epidural space is exposed to minimize the risks of elevations in intracranial pressures and neurological deficits. Lowering of the irrigation pressure also may lead to difficulties in controlling bleeding.

Short- and medium-term studies have shown that favorable outcomes after endoscopic fusion surgery as compared to the traditional open and minimally invasive tubular techniques. In fact, there is evidence that unilateral biportal lumbar endoscopic interbody fusion has significantly better short-term outcomes (in terms of smaller incision, less bleeding and shorter hospital stay) compared to MT-TLIF techniques while having similar fusion rates [14,15]. This was attributed to reduced muscle injury during the approach for the procedures.

The additional theoretical benefits of the minimally invasive nature of the surgery and preservation of the normal anatomy, as well as the long term fusion rates with better endplate preparation and reduced risks of endplate fractures/subsidence, require a closer look in the future.

## 6. Conclusions

The unilateral biportal lumbar endoscopic interbody fusion technique is an excellent technique in performing fusion while also minimizing complications of nerve injury, assisting in preparation of the endplate to aid fusion as well as preserving the normal anatomy of the spine. Patients can often be discharged the next or following day with significant improvements in their symptoms. This technique would be a good tool in the armamentarium of a spinal surgeon specializing in minimally invasive spinal surgery.

**Author Contributions:** Data curation, P.H.W.; Formal analysis, P.H.W.; Investigation, P.H.W.; Methodology, E.T.-C.L. and P.H.W.; Validation, E.T.-C.L.; Writing—original draft, E.T.-C.L. and P.H.W.; Writing—review & editing, E.T.-C.L. and P.H.W. All authors have read and agreed to the published version of the manuscript.

**Funding:** The research received no external funding.

**Informed Consent Statement:** Informed consent was obtained in all patients whose clinical photograph, radiograph and data presented.

**Data Availability Statement:** Data is available with corresponding authors through direct contact.

**Acknowledgments:** We would like to acknowledge Rajeesh George for providing assistance in the write-up of this paper.

**Conflicts of Interest:** Pang-Hung Wu as first co-author declared his spouse is the director of Singapore-based company Endocare PTE Ltd., which distributes orthopedic and spine products, including BESS, NSK drills and Bonss energy systems. No other co-authors have conflict of interest. No benefits in any form have been or will be received from a commercial party related directly or indirectly to the subject of this manuscript.

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
