# Peer review of "Technical Note on Unilateral Biportal Lumbar Endoscopic Interbody Fusion"

_2038-9582, doi:10.3390/std11020007_

Round 1

Reviewer 1 Report

Overall this is an interesting paper on utilizing a biportal technique to achieve a minimally invasive TLIF.  This is potentially a technique which may draw more surgeons into utilizing this approach compared to single portal techniques by allowing more traditional instruments and techniques used in open surgery

Author Response

Thank you so much for your positive feedback. 

Reviewer 2 Report

Overall, a well presented presentation of an endoscopic surgical technique for endoscopic intervertebral fusion. The basic procedure is not new, but the detailed approach shows some peculiarities which, to my knowledge, have not been published in this form before. The work could be published in its present form; I only recommend that the following additions be considered for improved comprehensibility: 

As a starting point, reference is made to the Kambin’s triangle, here a graphical representation would be useful for visual orientation.

Graphs 2-6 show good intraoperative representations. For readers who are not familiar with this surgical procedure, a correlating schematic representation would be very helpful to better understand the photographic documentation. I do not see this as absolutely necessary, but it should be considered in order to reach a wide readership.

Author Response

Thank you for your review and comments.

We included Kambin's triangle in figure 1.

 We had described in detail the steps with intraoperative photographs. Although no schematic diagram was provided due to limitations in the funding, figure 2 highlights the resection points for fusion. We hope you can accept our limitations. Thank you so much. 

Reviewer 3 Report

In this paper, the authors described the steps and technical pearls pertaining to unilateral biportal lumbar endoscopic interbody fusion and methods to avoid the common pitfalls and complications and they concluded this technique would be a good tool in the armamentarium of a spinal surgeon specializing in minimally invasive spinal surgery The methods are well designed and results are convincing. The conclusions of this study were supported fully by data. There were only two questions to be addressed:

1.In case1, we noticed the authors used routine interbody cage and in case 2 and 3 used special expandable interbody cage, and what is the differences between two types of interbody cage? How to choose them? 

2.All the cages used seem to be a little large, since we always used smaller cages and put allogeneic bone fragment between intervertebral space in front of cages, which is different with the method this manuscript descried. What are the considerations when evaluating these two methods?

Author Response

Thank you so much for your review.

1) We decide routine bullet interbody cage in patients who are osteoporotic ie T score is >-2.5 while the expandable cage is used when the patient is not osteoporotic. The reason is that we believe an expandable cage can create extra stress on the endplates leading to subsidence. 

2) Biportal Endoscopic Fusion resects facet completely and hence able to insert a large cage and packed as much bone graft as possible. The size of cage is not limited as the working portal comes from a separate incision. 

We had included point 1 and 2 in the article discussion second paragraph